# Oxidative Stress and Antioxidant Status in Pregnant Women with Gestational Diabetes Mellitus and Late-Onset Complication of Pre-Eclampsia

**DOI:** 10.3390/ijms26083605

**Published:** 2025-04-11

**Authors:** Kamelia Petkova-Parlapanska, Denitsa Kostadinova-Slavova, Mariya Angelova, Rafaah Sadi J. Al-Dahwi, Ekaterina Georgieva, Petya Goycheva, Yanka Karamalakova, Galina Nikolova

**Affiliations:** 1Department of Chemistry and Biochemistry, Medical Faculty, Trakia University, 11 Armeiska Str., 6000 Stara Zagora, Bulgaria; kamelia.parlapanska@trakia-uni.bg; 2Obstetrics and Gynaecology Clinic, UMHAT “Prof. St. Kirkovich”, 6000 Stara Zagora, Bulgaria; denitsa.kostadinova@trakia-uni.bg (D.K.-S.); rafa.dzhasim@trakia-uni.bg (R.S.J.A.-D.); 3Department of Obstetrics and Gynecology, Faculty of Medicine, Trakia University, 6000 Stara Zagora, Bulgaria; 4Department of General and Clinical Pathology, Forensic Medicine, Deontology and Dermatovenerology, Medical Faculty, Trakia University, 11 Armeiska Str., 6000 Stara Zagora, Bulgaria; ekaterina.georgieva@trakia-uni.bg; 5Propaedeutic of Internal Diseases Department, Medical Faculty, Trakia University Hospital, 6000 Stara Zagora, Bulgaria; petya.goycheva@trakia-uni.bg

**Keywords:** oxidative stress, gestational diabetes mellitus, NO, pre-eclampsia, eNOS, iNOS

## Abstract

Oxidative stress is a critical factor in the onset of gestational diabetes and its associated complication, pre-eclampsia. This study aimed to evaluate (1) reactive oxygen species, reactive nitrogen species, and superoxide radical levels as indicators of oxidative stress, (2) lipid and protein oxidation, (3) antioxidant enzyme activity, and (4) cytokine production in pregnant women with gestational diabetes, as well as those with both gestational diabetes and pre-eclampsia, comparing these with biomarkers of gestational diabetes mellitus. The study categorized pregnant patients with gestational diabetes mellitus into two groups based on the presence of new-onset hypertension, measured twice every four hours, and a 24 h urine protein test showing 300 mg/day or ≥1+ proteinuria detected via a visual dipstick at ≥20 weeks of gestation. These groups were compared with normotensive pregnant patients. The findings revealed that patients with both gestational diabetes and pre-eclampsia exhibited significantly elevated levels of reactive oxygen species, cytokine production, and lipid and protein oxidation end products compared to normotensive pregnant women. Additionally, these patients showed reduced nitric oxide (•NO) levels, impaired NO synthase systems (eNOS and iNOS), and decreased antioxidant enzyme activities (*p* < 0.05). These results indicate that patients with gestational diabetes and pre-eclampsia are unable to counteract oxidative stress effectively. The study underscores the compromised oxidative status as a contributing factor to these complications. The findings provide insights into the pathogenesis of gestational diabetes and the subsequent pre-eclampsia, the role of oxidative stress, and the resulting complications. Measuring oxidative stress levels and inflammatory biomarkers could help in the early detection and prediction of gestational-diabetes-related complications in pregnant women.

## 1. Introduction

Gestational diabetes (GD) and pre-eclampsia (PE) are major complications of pregnancy with significant clinical and societal impacts. Pre-eclampsia affects 2–15% of pregnancies and is characterized by new-onset hypertension after 20 weeks of gestation, accompanied by proteinuria or systemic organ dysfunction [1,2]. This condition threatens both maternal and fetal health, contributing to adverse outcomes such as fetal growth restriction, preterm birth, and maternal morbidity. Despite its prevalence, the molecular mechanisms underlying PE remain poorly understood, although placental oxidative stress has been implicated as a key driver [3,4,5]. Elevated markers of oxidative damage (e.g., oxidized lipids and proteins) are observed in PE, suggesting that placental oxidative stress overcomes endogenous antioxidant defenses, leading to systemic inflammation and endothelial dysfunction [6]. This oxidative imbalance may also impair insulin signaling, linking PE to gestational diabetes (GD). Gestational diabetes further exacerbates pregnancy risks, including macrosomia, stillbirth, and neonatal hypoglycemia [7,8]. However, 13% of women fail to achieve glycemic goals, and current therapies focus on glucose control without addressing oxidative stress or inflammation [9]. The co-occurrence of GD and PE significantly increases the risks of fetal abnormalities, cesarean delivery, preterm birth, and low birth weight [10,11,12,13]. Despite these associations, few studies have comprehensively examined the markers of oxidative stress (e.g., reactive oxygen/nitrogen species, and lipid/protein oxidation), antioxidant enzyme activity, and inflammatory cytokines in the progression of GD and PE. The study aims to monitor (1) the levels of reactive oxygen species, reactive nitrogen species, and superoxide radical levels as parameters of oxidative stress, (2) lipid and protein oxidation, (3) antioxidant enzyme activity, and (4) cytokine production in patients with gestational diabetes and pregnant women with gestational diabetes and pre-eclampsia and compare them with biomarkers for gestational diabetes mellitus. The novelty of the present study is that it is the first comprehensive profiling of oxidative/inflammatory pathways in the comorbidity of GD and PE. It identifies potential mechanistic links between oxidative stress, GD severity, and the onset of PE. It provides a basis for novel therapeutic strategies targeting oxidative stress (e.g., antioxidant therapies) in conjunction with glycemic control.

## 2. Results

### 2.1. Antioxidant Enzyme Activities (SOD, CAT, and GPx)

The activities of antioxidant enzymes, specifically catalase (CAT), superoxide dismutase (SOD), and glutathione peroxidase (GPx), showed statistically significant changes when compared to normal pregnancies. The antioxidant system activity in GDM and GDM + PE was significantly depleted. The activity of SOD in GDM + PE was statistically considerably decreased compared to controls (*p* = 0.001) and compared to GDM (*p* = 0.02). The SOD activity in GDM showed similar results compare to controls. The results indicated an insignificant difference in CAT activity between the GDM (*p* = 0.000128) and GDM + PE (*p* = 0.00013), (Table 1). The CAT activity was significantly higher in GDM (*p* = 0.0001) and GDM + PE (*p* = 0.0001) compared to controls. The analysis of results indicated that groups of patients, GDM (0.00014) and GDM + PE (0.00011), had significantly lower GPx activity compared to controls (*p* = 0.0001). The analysis of these results indicates that the reduced activity of these antioxidant enzymes is due to their depletion (*p* < 0.00). A woman’s high weight before pregnancy significantly affects the risk of complications during pregnancy, such as gestational diabetes mellitus (GDM) and hypertension, which can lead to adverse outcomes for the fetus.

### 2.2. Gestational Diabetes Mellitus Biomarkers

Serum adiponectin (APN) levels were significantly lower (*p* = 0.004) in pregnant women with gestational diabetes mellitus (GDM) compared to those without GDM (see Table 2). The group of pregnant women with GDM had higher leptin levels (*p* = 0.00) compared to normotensive pregnant women. C-reactive protein (CRP) levels were significantly higher in the GDM group than in the normal group (*p* = 0.005). Table 2 shows the values of APN, leptin, and C-reactive protein (CRP) in the group of pregnant women with gestational diabetes complicated by pre-eclampsia (GDM + PE), and levels of the biomarkers are statistically significantly increased vs. controls and similar to GDM. Serum advanced oxidation protein product (AOPP) levels in pregnant women with GDM and PE were statistically elevated and correlated with the simultaneous onset of GDM and PE.

### 2.3. Levels of Protein Carbonyls (PCs) and Advanced Glycation End Products (AGEs)

In the groups of pregnant women with gestational diabetes mellitus (GDM) and those with GDM complicated by pre-eclampsia (GDM + PE), no statistically significant difference (*p* = 0.05) was found in the levels of advanced glycation end products (AGEs) (Figure 1A). However, both groups exhibited a statistically significant increase in AGEs compared to the normal pregnancy (controls) group (*p* < 0.05).

The levels of protein carbonyls (Figure 1B) in pregnant patients with GDM and GDM *+* PE complications were significantly higher compared to the control group (*p* < 0.05). No significant difference was observed between the two groups of GDM patients when compared to the GDM *+* PE group (*p* = 0.00). A positive correlation was found between AGEs (pg/mL) and PC levels (r = 0.69; *p* ≤ 0.00).

### 2.4. Assessment of Oxidative Stress by Measuring the ROS, MDA, and 8-Iso-Pr

During the third trimester, insulin resistance and an increase in lipid metabolism led to elevated reactive oxygen species (ROS) production (Figure 2A). Both the gestational diabetes mellitus (GDM) and GDM + pre-eclampsia (PE) groups exhibited statistically significant differences in ROS levels (*p* = 0.01) and had higher ROS levels compared to the controls (*p* = 0.05) (Figure 2A).

Similar results were observed for the levels of malondialdehyde (MDA) (Figure 2B) and 8-iso-prostaglandin (Figure 2C). Notably, the GDM + PE group showed statistically higher levels compared to the GDM group (*p* < 0.05), and both groups reported significantly increased levels compared to the control group (*p* < 0.05).

### 2.5. The Levels of DNA Damage

The quantification of 8-OHdG (Figure 3) was measured in blood and was statistically significantly increased compared to controls in both the GDM group (11.02 ± 1.2 [pg/mL] vs. 4.59 ± 0.81 [pg/mL], *t*-test *p* < 0.05) and GDM + PE (17.64 ± 3.5 [pg/mL] vs. 4.59 ± 0.81 [pg/mL], *t*-test *p* < 0.05). A statistically significant difference was also reported in 8-OHdG levels in the GDM + PE group compared to GDM (*p* = 0.00).

### 2.6. Assessment of Oxidative Stress by Measuring Nitrosative Stress

From the EPR studies presented in Figure 4A, depleted •NO levels are reported in the GDM + PE groups compared to GDM (*p* < 0.05) (Figure 4A). A statistically significant decrease is reported in both groups, GDM (*p* < 0.05) and GDM + PE (*p* < 0.05), compared to NP. The results presented for iNOS (Figure 4B) are similar. Figure 4C present a statistically significant difference in eNOS in the group with GDM + PE compared to GDM (*p* < 0.05). The measured superoxide anion radical is increased many times in both the GDM (*p* < 0.05) and GDM + PE (*p* < 0.05) groups compared to healthy pregnant women (Figure 4D). There is no statistically significant difference between GDM and GDM + PE (*p* < 0.00).

### 2.7. The Levels of Pro-Inflammatory Cytokines

The level of interleukin 1α (IL-1α; Figure 5A) in the GDM + PE women was statistically significantly higher compared to the NP (LSD post hoc test, *p* < 0.05) and GDM group (*p* < 0.05). Interleukin 1β levels (IL-1β; Figure 5B) were statistically significantly decreased compared to controls (*p* < 0.05) and the GDM group (*p* < 0.05). In the GDM + PE group, interleukin 6 (IL-6) levels (Figure 5C) were statistically significantly increased both in comparison to controls (LSD post hoc test, *p* < 0.05) and in contrast to GDM (LSD post hoc test, *p* < 0.05). In the two groups with complications, GDM and GDM + PE, no statistically significant difference was reported (LSD post hoc test, *p* = 0.1). Interleukin 10 (IL-10; Figure 5D) was statistically significantly decreased in the GDM + PE group both as sleep controls (*p* < 0.05) and relative to GDM (*p* < 0.05). Interleukin 17β (IL-17; Figure 5E) showed a statistical difference between the GDM + PE and the control group. The GDM + PE group was statistically significantly higher compared to controls (*p* < 0.05) and GDM (*p* < 0.05). The interferon gamma results (INF-γ; Figure 5E) showed a statistically significant increase in the GDM + PE group compared to controls (*p* < 0.05) and against the GDM group (*p* < 0.05). The mean value of the tumor necrosis factor (TNF-α; Figure 5G) in the GDM + PE group was statistically significantly higher compared to the controls (post hoc test, *p* < 0.05) and to the group with GDM (*p* < 0.05). Transforming growth factor-β (TGF-β; Figure 5H) also showed a statistically significant increase in the GDM + PE group compared to controls (*p* < 0.05) and against the group with gestational diabetes (*p* < 0.05).

A positive correlation is observed between insulin levels (uU/mL) and PC (r = 0.43, *p* = 0.00), AGEs (r =0.81, *p* = 0.00), 8-IsoPr (r = 0.83, *p* = 0.00), MDA (r = 0.87, *p* = 0.00), •O_2_^−^ (r = 0.74, *p* = 0.00), IL-17A (r = 0.84, *p* = 0.00), IL-10 (r = 0.76, *p* = 0.00), TGF-α (r = 0.89, *p* = 0.00), TNF-α (r = 0.93, *p* = 0.00), and INF-γ (0.93, *p* = 0.00); and a negative correlation is observed with NO (r = −0.89, *p* = 0.00), eNOS (r = −0.92, *p* = 0.00), iNOS (r = −0.91, *p* = 0.00), IL-1β (r = −0.79, *p* = 0.00), and IL-10 (r = −0.92, *p* = 0.00).

## 3. Discussion

In placentas impacted by pre-eclampsia, the expression of Nrf2 rises substantially as a compensatory response to oxidative stress. However, the activity of antioxidant enzymes like superoxide dismutase (SOD) and glutathione peroxidase (GSH-Px) is significantly diminished, and the catalase (CAT) is statistically significantly increased. Elevated circulating levels of advanced glycation end products (AGEs) are correlated with oxidative stress and inflammation, both implicated in pre-eclampsia development. Research indicates that AGEs act as biomarkers for pre-eclampsia exclusively in cases where gestational diabetes mellitus (GDM) coexists [14]. Notably, no marked differences in AGE levels were observed between patients with GDM alone and those with both GDM and pre-eclampsia (GDM + PE). AGEs facilitate the cross-linking of long-chain proteins, exacerbating microvascular complications [15,16,17]. Adiponectin (APN), a hormone vital in glucose and lipid metabolism regulation, is predominantly synthesized by adipose tissue adipocytes and exerts systemic effects, including hepatic modulation [18]. Reduced APN levels correlate with heightened insulin resistance, elevated blood glucose, and impaired insulin sensitivity. These findings, consistent with multiple studies [19,20,21], align with the data in Table 2. Research further demonstrates that serum APN levels are markedly lower in GDM patients compared to non-GDM individuals, underscoring its utility as a biomarker for glycemic monitoring during pregnancy [22]. Leptin interacts with pancreatic β-cell receptors, suppressing insulin secretion in response to glucose and subsequently raising blood glucose levels. Elevated leptin concentrations are associated with GDM, as evidenced by higher serum leptin levels in GDM patients [23], a finding corroborated by Table 2 and other studies. For instance, Plowden et al. [24] highlighted that increased leptin levels in pregnancy correlate with adverse GDM outcomes. C-reactive protein (CRP) is closely tied to GDM pathogenesis, with elevated early-pregnancy CRP levels heightening GDM risk [25], promoting it as a potential predictive marker for the condition.

Conditions such as obesity, diabetes, and pre-eclampsia trigger stress via multiple pathways. Key mechanisms involve the production of superoxide anions, activation of protein kinase C, hyperglycemia, elevated cholesterol and triglyceride levels, mitochondrial impairment, reduced antioxidant defenses, chronic inflammation, and reactive oxygen species (ROS) generation [26]. Correlation studies indicate a positive link between advanced glycation end products (AGEs) and protein carbonyls (PCs). The aggregation of extensively cross-linked proteins fosters oxidant formation, damaging cellular macromolecules and culminating in apoptotic cell death [27]. Carbonyl proteins arise either through the direct oxidation of amino acids like proline, arginine, lysine, and threonine, or via secondary interactions of cysteine, histidine, and lysine with reactive carbonyl compounds (e.g., aldehydes and ketones) derived from lipid peroxidation or glycation/glycoxidation [28]. A notable finding is the lack of significant statistical differences in protein carbonyl levels between pregnant women with gestational diabetes and those with both gestational diabetes and pre-eclampsia. This observation likely explains the reported results, as carbonyl compounds are precursors to carbonyl stress, exacerbating oxidative damage and cellular dysfunction [29,30].

Under hyperglycemic conditions, ROS and AGEs are linked to impaired •NO function and excessive production of vasoconstrictors. Prior research corroborates diminished •NO bioavailability and/or heightened ROS generation, disrupting the balance between •NO and superoxide (O₂•⁻). This imbalance promotes peroxynitrite (ONOO⁻) formation and suppresses endothelial nitric oxide synthase (eNOS) activity [31]. Elevated O₂•⁻ levels accelerate NO degradation and ONOO⁻ synthesis. Notably, ONOO⁻ induces protein nitration, compromising vascular regulators like nitric oxide synthase (NOS) [32]. While earlier studies estimated •NO levels indirectly via nitrite/nitrate measurements (e.g., Griess method), the present study employs electron paramagnetic resonance (EPR) with the carboxy-PTIO spin trap. This method captures stable NO adducts, enhancing measurement precision (Figure 4A) [33]. The data align with evidence implicating oxidative stress in pre-eclampsia pathogenesis [34]. Specifically, reduced plasma •NO levels in gestational diabetes patients may exacerbate pre-eclampsia progression, as confirmed by the significantly lower •NO in individuals with both gestational diabetes and pre-eclampsia. Placental ischemia further diminishes the antioxidant capacity, amplifying oxidative stress. This cascade drives pathological features of pre-eclampsia, such as hypertension and proteinuria, by overwhelming antioxidant defenses and perpetuating macromolecular damage [35].

Oxidative damage to lipids and proteins is recognized as a contributor to insulin resistance and subsequent complications. In GDM, maternal circulating levels of oxidative stress markers—such as MDA, reactive substances, and lipid hydroperoxides (LOOHs)—are significantly elevated compared to normal pregnancies [36]. These biomarkers, particularly MDA and lipid peroxides, are also increased in placental tissue from PE cases relative to healthy pregnancies [31]. Notably, plasma MDA levels rise further in PE patients at 18–22 and 26–28 gestational weeks [31], a trend corroborated by our findings showing markedly higher MDA levels in both groups with GDM and GDM + PE. Sande et al. [37] highlighted a stronger association between type 1/2 diabetes and premature PE onset. Our study aligns with this, as patients with confirmed GDM developed PE earlier in gestation. Recent data further reveal a positive correlation between MDA, ROS, and NO levels. Intriguingly, GDM + PE patients exhibit elevated CAT activity alongside reduced SOD and GPx activity. This imbalance disrupts redox homeostasis during the third trimester, exacerbating lipid peroxidation (evidenced by the GSH/GSSG ratio).

Alarmingly, studies on GDM-PE pregnancies consistently report depleted activities of key antioxidant enzymes (CAT, SOD, and GPx), a pattern confirmed by multiple investigations [38]. These findings underscore that GDM is closely tied to heightened oxidative stress and inflammatory dysregulation, emphasizing the critical need for targeted research and clinical interventions to mitigate these risks.

Research highlights that pregnant women with gestational diabetes mellitus (GDM) often exhibit increased lipid levels across all trimesters [39]. However, LDL cholesterol shows a notable decline during the second and third trimesters. These lipid abnormalities influence both the amount and composition of lipids passed to the fetus, posing heightened risks to newborn health as pregnancy advances. Notably, lipid and triglyceride levels in the third trimester are more closely linked to birth weight than glucose levels. In GDM cases with elevated lipids, reduced inflammatory markers—specifically IL-1β (Figure 5B) and TNF-α (Figure 5G)—are observed in maternal blood and placental tissue, potentially tied to enhanced fatty acid synthesis [40]. Further, placental phospholipids in GDM display higher concentrations of arachidonic acid (AA) and docosahexaenoic acid (DHA), alongside upregulated lipid-related gene activity, suggesting adaptations in fetal–placental pathways that may amplify lipid transfer, particularly in GDM-complicated pregnancies. Hormonal factors like progesterone and glucocorticoids exacerbate insulin resistance by altering maternal glucose and lipid metabolism. In typical pregnancies, inadequate β-cell function can impair carbohydrate tolerance [41]. Together, insulin resistance and insufficiency elevate maternal glucose, amino acids, and lipids, driving the development of GDM (see Table 1).

Oxidative stress is a natural aspect of pregnancy, arising from placental mitochondrial function, ROS, and metabolic byproducts of cellular processes [42]. Key endogenous ROS sources include mitochondria, endoplasmic reticulum, and peroxisomes. Elevated ROS levels trigger the release of damaging mediators into maternal circulation, often observed in cases of poor placental development and subsequent placental ischemia. Placental insufficiency, linked to complications like PE and intrauterine growth restriction, is hypothesized to originate from the inadequate trophoblast invasion of uterine spiral arteries. This disrupts uteroplacental blood flow, causing impaired placentation, ischemia, oxidative stress, inflammation, and syncytiotrophoblast apoptosis. Maternal obesity or gestational diabetes exacerbates these pathways, promoting visceral fat accumulation, adipocyte dysfunction, and heightened ROS production, which further amplifies insulin resistance in adipose and peripheral tissues [43].

Excessive ROS generation critically influences vascular responses and apoptosis [43]. The ROS-induced oxidation of the IκB kinase complex activates nuclear factor kappa B (NF-κB), driving the transcription of proinflammatory cytokines (e.g., IL-6, TNF-α; Figure 5B,G) and mediators of endothelial dysfunction. While this interaction is tightly regulated in healthy pregnancies, it becomes dysregulated in GDM and GDM with PE. Nitric oxide, essential for vascular homeostasis and tone, is disrupted by ROS through reduced NO bioavailability and the suppression of eNOS and iNOS [44]. These disruptions impair vascular tone regulation, platelet adhesion, and leukocyte activity. Diminished NO levels and synthase activity are tied to endothelial dysfunction in GDM and GDM + PE, reflecting faulty maternal vascular adaptation. During pregnancy, NO—produced by eNOS and iNOS—supports embryonic development, implantation, trophoblast invasion, and placental vascular function [45]. The ROS–NO balance is vital for maintaining vascular health, and its disruption can lead to adverse outcomes.

In pregnancies complicated by gestational diabetes mellitus and pre-eclampsia, Figure 4A demonstrates diminished NO levels alongside an elevated eNOS expression (Figure 4D) [46]. Dysregulated eNOS activity, a key enzyme in NO production, is linked to endothelial dysfunction caused by impaired maternal vascular adaptation. These observations align with findings by Espinoza et al. [47], who describe eNOS uncoupling—a phenomenon where increased eNOS expression in maternal and cord blood paradoxically correlates with reduced NO bioavailability. During uncoupling, eNOS dysfunction promotes monomerization, shifting its output from NO to superoxide radicals via peroxynitrite formation [48]. Additionally, excess reactive oxygen species (ROS), generated by mitochondrial dysfunction in endothelial cells under hypoxic or hyperglycemic conditions, further inactivate NO, exacerbating hypertension and vascular damage [49]. In healthy pregnancies, placental cytokine signaling supports vascular development, but this process is disrupted in PE, leading to obstetric complications. Histopathological studies reveal elevated pro-inflammatory cytokines (TNF-α and TGF-β1) and reduced anti-inflammatory IL-10 in PE patients. McElwain et al. [50] identify a distinct “cytokine signature” in PE, marked by heightened pro-inflammatory mediators (IL-8, IL-6, and IFN-γ) compared to normotensive pregnancies, with severe PE cases showing the highest levels. Uterine-derived IL-6 contributes to systemic endothelial dysfunction, amplifying vascular pathology. Adipose tissue releases factors such as leptin, adiponectin, resistin, TNF-α, IL-6, and IL-1β [51], most of which (except adiponectin) drive inflammation. Excess leptin reduces NO availability, increases oxidative stress, and promotes platelet activation, raising thrombosis risk.

## 4. Materials and Methods

### 4.1. Chemicals

All reagents were analytical-grade dimethyl sulfoxide DMSO (472301 CAS no. 472301); 2-thiobarbituric acid (T5500; CAS no. 504-17-6) were supplied by Merck (Sofia, Bulgaria), and N-tert-butyl-α-phenylnitrone, PBN (B7263; CAS no. 3376-24-7), and carboxy-PTIO potassium salt (C221; CAS no. 148819-94-7) were purchased from Sigma-Aldrich, Sofia, Bulgaria EAD.

### 4.2. Diagnostic Criteria for Outcomes

GDM was diagnosed when two or more plasma glucose values during the diagnostic OGTT met or exceeded the criteria for a positive test as recommended by the National Diabetes Data Group [plasma glucose cutoffs: fasting 5.8 mmol/L (105 mg/dL), 1 h 10.5 mmol/L (190 mg/dL), 2 h 9.1 mmol/L (165 mg/dL), and 3 h 8.0 mmol/L (145 mg/dL)]. Diet therapy with blood glucose monitoring was recommended for patients with GDM. Pre-eclampsia was characterized by new-onset hypertension measured twice every four hours (140/90) and a 24 h urine protein test of 300 mg/day at ≥20 weeks of gestation or ≥1+ proteinuria detected using a visual dipstick. PE was defined as the occurrence of hypertension during pregnancy: systolic blood pressure ≥140 mmHg and diastolic blood pressure ≥90 mmHg, plus albuminuria; or the presence of ≥300 mg of protein in a 24 h urine sample after 20 weeks of gestation in a woman who was normotensive before 20 weeks of gestation.

### 4.3. Patients

The present study comprised 30 patients with GDM, 28 patients with GDM and PE, and 27 subjects with normal pregnancy who were enrolled in regular checkups performed in the Department of Obstetrics and Gynecology Clinic, UMHAT “Prof. St. Kirkovich” Stara Zagora, Bulgaria, between April 2023 and August 2024. GDM + PE was considered when hypertension (≥140/90 mmHg) was detected on two separate occasions with an interval of >15 min (6) h for the first time in previously normotensive pregnant women at 20 weeks of gestation according to the International Society for the Study of Hypertension in Pregnancy (ISSPH) with involvement of one or more organs. Proteinuria was considered to be present when the 24 h total urinary protein excretion was ≥300 mg/24 h. GDM was diagnosed using the criteria of the American Diabetes Association (ADA), where, following a 75 g oral glucose tolerance test during 24–28 weeks of gestation, one of the three following categories was fulfilled: based on the one-step approach recommended by the ADA, the pregnant women were defined as having GDM if they had at least one abnormal high-glucose value out of three 75 g OGTT, fasting blood glucose (FBG), ≥5.1 mmol/L, 1 h post 75 g oral glucose load, ≥10.0 mmol/L, and 2 h post 75 g oral glucose load, ≥8.5 mmol/L. Patients with cardiovascular, renal, and metabolic disease and an abortion history were excluded from the study. None of the patients were undergoing treatment or received any medication within 2 months before sample collection. The ethics committee of the UMHAT, “Prof. St. Kirkovich” (Stara Zagora, Bulgaria), approved the study protocol, and all participants provided written informed consent, listed in Table 1.

### 4.4. Blood Samples Preparation

Fasting samples of venous blood from each patient and controls were collected in the morning between 07:00 and 09:00 h in clot blood tubes, centrifuged at 3000 rpm for 10 min at 5 °C, and serum was carefully separated and used immediately.

### 4.5. Electron Paramagnetic Resonance (EPR) Study

All electron paramagnetic resonance (EPR) measurements of all tested samples were conducted at room temperature (18–23 °C) on an X-band EMXmicro, spectrometer Bruker, Bremen, Germany, equipped with standard Resonator. Quartz capillaries were used as sample tubes. The sample tube was sealed and placed in a standard EPR quartz tube (i.d. 3 mm) which was fixed in the EPR cavity. All EPR experiments were carried out in triplicate and repeated. Spectral processing was performed using Bruker, WIN-EPR version 2021 and SimFonia software version 2021.

#### 4.5.1. EPR Evaluation of ROS Product

Preparation of the sera samples and EPR study of the ROS production were performed according to Shi et al. [52]. Briefly, to 0.1 mL of sera samples were added 1.0 mL of a 50 mM solution of the spin-trapping agent PBN dissolved in DMSO. After centrifugation, a 0.4 mL sample was taken in the quartz tube for measurement. EPR spectra were recorded at room temperature. EPR settings were as follows: center field 3503.74 G, microwave power 20.42 mW, modulation amplitude 0.50 G, sweep width 100 G, gain 1 × 10^6^, time constant 327.68 ms, sweep time 81.92 s, and 5 scans per sample.

#### 4.5.2. EPR Evaluation of •NO Radical

Based on the previous methods [53,54], we developed and adapted the EPR method for evaluation of the levels of radical •NO radicals. Briefly, the solution of Carboxy. PTIO.K (50 μM) was prepared after dissolving in a mixture of Tris buffer (50 mM, pH 7.5) and DMSO in a ratio 9:1. To 100 μL plasma/brain homogenates was added 900 μL Tris buffer plus DMSO (9:1) and centrifuged at 4000 rpm for 10 min at 4 °C. The tested sample (100 μL) and 100 μL 50 μM solution of Carboxy. PTIO were mixed. The EPR spectrum of the spin adduct formed between the spin trap Carboxy. PTIO and generated radical •NO radicals was recorded. The levels of radical dot NO radicals were calculated as double integrated plots of EPR spectra and results were expressed in arbitrary units. The EPR settings were as follows: 3505 G centerfield, 6.42 mW microwave power, 5 G modulation amplitude, 75 G sweep width, 2.5 × 10^2^ gain, 40.96 ms time constant, 60.42 s sweep time, and 1 scan per sample.

#### 4.5.3. EPR Evaluation of Superoxide Anion Radical

The serum was incubated for 30 min at 37 °C with a superoxide-sensitive spin probe, 1-hydroxyl-3-methoxycarbonyl-2,2,5,5-tetramethylpyrrolidine (CMH), and immediately measured. The amplitude of the spectrum is directly proportional to the •O_2_^−^ concentration [55].

### 4.6. Enzyme-Linked Immunosorbent Assay

All markers of oxidative stress were measured with an ELISA reader (Multiska FC, Microplate Photometer, Thermo Scientific, Karlsruhe, Germany), following the manufacturer’s instructions. The ELISA kits were as follows: Human Malondialdehyde MDA (ab233471), Human Protein Carbonyl ELISA Kit (ab238536), Human AGEs levels ELISA Kit (ab238539), Human Superoxide Dismutase SOD (ab119520), Human Catalase CAT (ab190523), Human Glutathione Peroxidase GPx (Cayman 703102), 8-OHdG (ab285254), Human eNOS (ab263878), and Human iNOS (ab253878).

### 4.7. Statistical Analysis

All experiments were performed in triplicate, and results are expressed as the mean ± SEM (Standard Error of the Mean). Statistical analyses were conducted using one-way ANOVA, 2008, followed by Tukey post hoc test. GraphPad Prism software 10.1.2 was used for all statistical calculations. Statistical significance was set at *p* < 0.05.

## 5. Conclusions

The article focuses on the study of the markers of oxidative stress and redox status in pregnant women with complications and their comparison with those of normotensive pregnant women. The measurement of ROS and RNS levels, lipid peroxidation products, DNA damage, antioxidant enzyme activity and inflammation, and the study of the correlation of these parameters will allow the assessment of the progression of the condition. Strategies involving the monitoring of redox status in patients with gestational diabetes and complications such as pre-eclampsia will help in the management of complications.

The presented study on gestational diabetes and gestational diabetes with pre-eclampsia can be summarized as follows:Comprehensive profiling of oxidative/inflammatory pathways in the comorbidity of gestational diabetes with pre-eclampsia. While oxidative stress and inflammation have been studied separately in GDM + PE, the present study is the first to systematically analyze these pathways in their co-occurrence. The study provides new insights into how oxidative stress and inflammation may interact to worsen pregnancy outcomes;Identification of mechanistic links between oxidative stress, GDM severity, and the onset of complications with subsequent PE. The study investigates whether oxidative stress in GDM contributes to the development of PE, filling a gap in the understanding of their shared pathophysiology. An analysis of the results provides a multiparametric assessment, not previously carried out in comorbid GDM-PE cases, by measuring the reactive oxygen/nitrogen species (ROS/RNS), lipid/protein oxidation, antioxidant enzyme activity, and cytokines;Potential for new therapeutic strategies. The current management of GDM focuses on glucose control, ignoring oxidative stress and inflammation. The findings presented could support antioxidant-based therapies (e.g., α-lipoic acid, vitamins C/E, and N-acetylcysteine), together with glycemic control, suggesting a new paradigm for the treatment of high-risk pregnancies;Bridging clinical and molecular gaps. Previous studies have linked GDM + PE epidemiologically, but few have profiled the biomarkers at different stages of the disease. By comparing patients with GDM alone and patients with GDM+PE, the study elucidates whether oxidative/inflammatory markers predict the risk of PE in women with GDM;Potential for early risk stratification. If specific oxidative/inflammatory signatures are associated with worse outcomes, they could serve as early diagnostic or prognostic markers to improve prenatal care. The present study may change clinical management by highlighting oxidative stress as a modifiable risk factor in high-risk pregnancies.

## Figures and Tables

**Figure 1 ijms-26-03605-f001:**
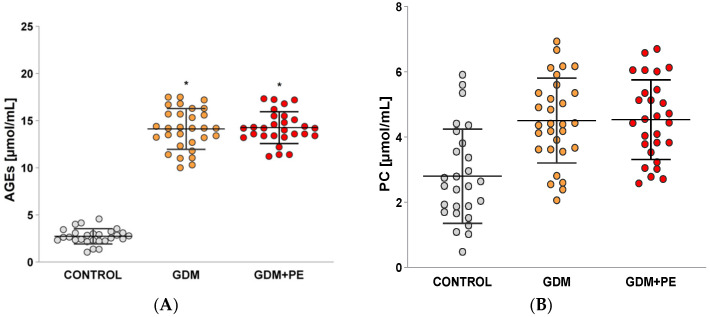
Shows the levels of oxidative stress markers presented as; levels of advanced glycation end product (AGE) controls, and protein carbonyl (PCC) levels: (**A**) AGEs: normotensive pregnancy (control); GDM patients; and GDM + PE; and (**B**) PCC: normotensive pregnancy (control); GDM patients; and GDM + PE; Turkey HSD post hoc test; * *p* < 0.05 vs. control group;.

**Figure 2 ijms-26-03605-f002:**
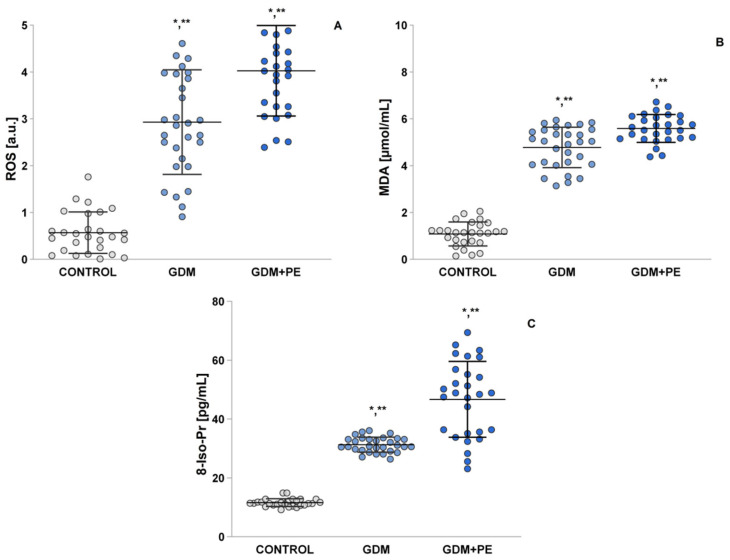
Shows the levels of oxidative stress markers presented as ROS production, MDA, and 8-Iso-prostaglandin: (**A**) ROS production: normotensive pregnancy (control); GDM patients; and GDM + PE; (**B**) MDA: normotensive pregnancy (control); GDM patients; and GDM + PE; and (**C**) 8-Iso-prostaglandin: normotensive pregnancy (control); GDM patients; and GDM + PE; Turkey HSD post hoc test; * *p* < 0.05 vs. control group; ** *p* < 0.05 vs. GDM + PE group.

**Figure 3 ijms-26-03605-f003:**
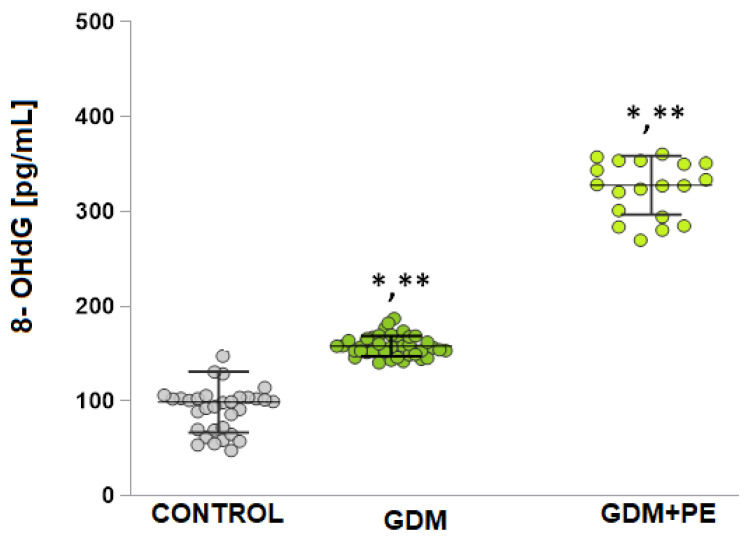
Shows the levels of damage in DNA presented as levels of 8-OHdG [pg/mL], measured in groups with the following: normotensive pregnancy (control); GDM patients; and GDM + PE. Turkey HSD post hoc test; * *p* < 0.05 vs. control group; ** *p* < 0.05 vs. GDM + PE group.

**Figure 4 ijms-26-03605-f004:**
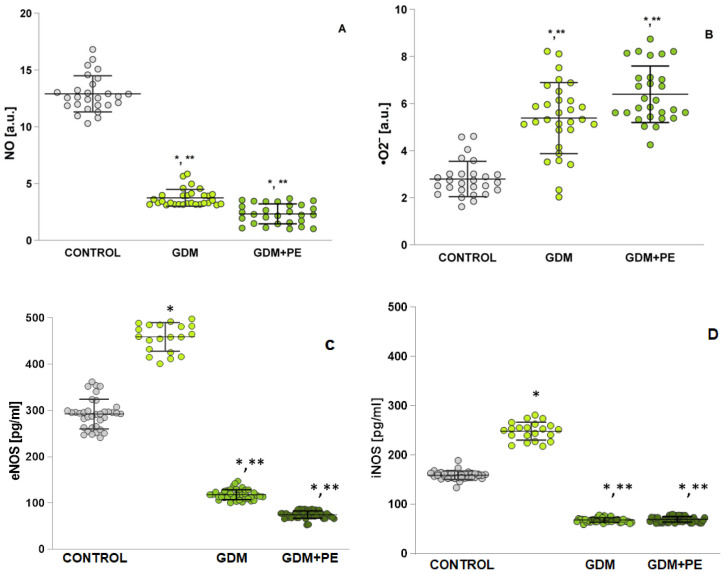
Shows the evaluation of the levels of oxidative stress markers presented as NO levels, •O_2‾_, eNOS, and iNOS in serum samples: (**A**) NO: normotensive pregnancy (control); GDM patients; and GDM + PE; (**B**) •O_2‾_: normotensive pregnancy (control); GDM patients; and GDM + PE; (**C**) eNOS: normotensive pregnancy (control); GDM patients; and GDM + PE; and (**D**) iNOS: normotensive pregnancy (control); GDM patients; and GDM + PE; Turkey HSD post hoc test; * *p* < 0.05 vs. control group; ** *p* < 0.05 vs. GDM + PE group.

**Figure 5 ijms-26-03605-f005:**
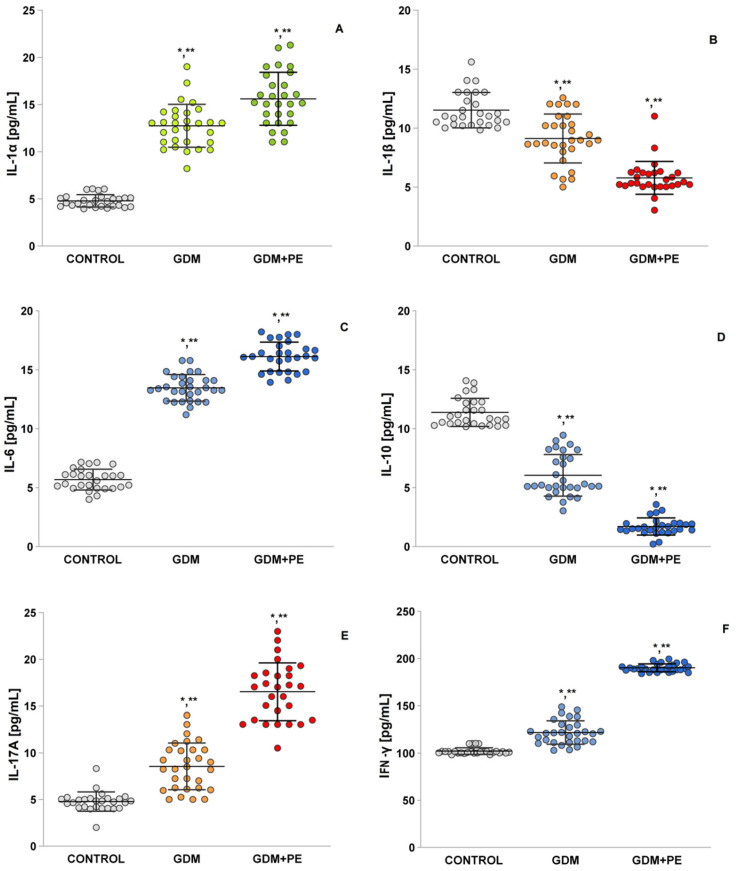
Pro-inflammatory cytokine levels: (**A**) IL-1α; (**B**) IL-1β; (**C**) IL-6; (**D**) IL-10; (**E**) IL-17A; (**F**) IFN-γ; (**G**) TNF-α; and (**H**) TGF-β; Tukey HSD post hoc test; * *p* < 0.05 vs. control group; ** *p* < 0.05 vs. GDM + PE group.

**Table 1 ijms-26-03605-t001:** Clinical and laboratory data of the patients divided into two groups (group with gestational diabetes mellitus (GDM) and group of pregnant women with gestational diabetes mellitus complicated with pre-eclampsia (GDM + PE)) depending on blood sugar levels (mmol/L), blood pressure (mmHg), and proteinuria (mg/dL/day). The results were compared with data for normotensive pregnant women (controls).

Variables	Normal Pregnancies (Controls) (*n* = 27)	GDM (*n* = 30)	GDM + PE (*n* = 28)
**Age (years)**	27.33 ± 9.28	31.21 ± 9.23	30.25 ± 8.14
**Gestational age at sampling (weeks)**	20–40	24–28	24–28
**BMI (kg/m²), mean ± SD**	25.23 ± 3.52	30.82 ± 4.26	31.52 ± 5.89
**BMI > 30 kg/m², *n* (%)**	7.4%	59.2%	58.1%
**Blood sugar (mmol/L), mean ± SD**	4.97 ± 0.32	8.77 ± 0.75	9.52 ± 5.18
**HbA1_C_% (mmol/mol)**	5.09 ± 0.71	6.90 ± 1.07	7.78 ± 0.67
**Fasting glucose (mmol/L)**	4.72 ± 0.99	5.001 ± 0.56	4.98 ± 0.49
**1 h plasma glucose after Oral glucose tolerance test OGTTb (mmol/L)** **2 h plasma glucose after OGTT (mmol/L)**	5.62 ± 0.87 6.87 ± 0.87	10.06 ± 0.71 8.7 ± 0.85	10.80± 0.88 8.8 ± 0.70
**Insulin (uU/mL)**	15.33 ± 0.21	20.12 ± 0.34	25.10 ± 0.26
**Cholesterol (mmol/L)**	4.43 ± 0.76	5.47 ± 0.6	5.12 ± 1.38
**Triglycerides (mmol/L)**	1.52 ± 0.44	2.513± 0.17	2.43 ± 1.27
**HDL (mmol/L)**	1.01 ± 0.28	1.95 ± 0.13	1.25 ± 0.38
**LDL (mmol/L)**	2.32 ± 0.62	2.32 ± 0.15	2.86 ± 1.10
**sFlT-1/PIGF**	5.12 ± 3.54	38.07 ± 6.31	68.61 ± 12.43
**sFlT-1 (soluble thyrosin kinase)**	898 ± 1.25	4006 ± 154.01	4976 ± 115.32
**PIGF (placental growth factor)**	177 ± 14.13	103 ± 11.10	90 ± 11.96
**Blood pressure (mmHg) systolic** **diastolic**	120 80	120 80	140 90
**Proteinuria (mg/dL/day)**	≤150	≤150	≥300
**Estimated Birth weight (g)**	3000 ± 501	3800 ± 493	2750 ± 481
**SOD U/gHb**	121 ± 15.55	61.24 ± 18.15	50.14 ± 13.11
**CAT U/gHb**	48.59 ± 8.66	69.77 ± 7.58	72.69 ± 11.45
**GPx U/gHb**	289.87 ± 25.58	132.35 ± 15.61	94.28 ± 12.12

**Table 2 ijms-26-03605-t002:** Predictors of gestational diabetes mellitus (GDM), and concurrent GDM with PE.

Predictive Factor	Normal Pregnancies (Controls) (*n* = 27)	GDM (*n* = 30)	GDM + PE (*n* = 28)	*p*
Adiponectin (APN) pg/mL	76.25 ± 18.66	38.33 ± 14.13	39.52 ± 8.01	*p* = 0.004 vs. controls
Leptin pg/mL	55.32 ± 19.17	78.99 ± 14.97	79.14 ± 20.01	*p*= 0.001
Human CRP pg/mL	4.38 ± 0.21	7.47± 0.14	7.68 ± 0.26	*p* = 0.005
Advanced oxidation protein products (AOPP) ng/mL	68.59 ± 16.09	89.56 ± 36.52	86.24 ± 16.98	*p* = 0.005

## Data Availability

The data presented in this study are available upon request from the corresponding author.

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
