# Peer review of "Oxidative Stress and Antioxidant Status in Pregnant Women with Gestational Diabetes Mellitus and Late-Onset Complication of Pre-Eclampsia"

_ijms, 2025, doi:10.3390/ijms26083605_

Round 1
Reviewer 1 Report
Comments and Suggestions for Authors
Dear Authors.
The manuscript you submitted, "Oxidative stress and antioxidant status in pregnant women with gestational diabetes mellitus and late-onset complication of preeclampsia” attempts to evaluate oxidative and inflammatory processes in patients with gestational diabetes alone or with PE. It is an interesting line of work in an area of significant clinical impact. Unfortunately, this manuscript lacks an adequate analysis of the results obtained. Moreover, presenting the figures that show the main results must be corrected.
When reading the title and the introduction, the authors explain and develop why the study and the comparisons between the conditions of the pregnant patients are necessary. The experimental determinations are adequate to answer the questions generated and thus account for the objective of the manuscript. The introduction is agile and precise in the statement of ideas, and the title gives an account of the main aim of the manuscript.
In the methodology section, the techniques are well described in some points; in others, they are very briefly described, for example, in sections 4.4.1 and 4.4.2, which mention modifications but do not express them. Although all the methodologies have an associated reference, should the manuscript reader understand that the same equipment was used, please mention which equipment the authors used to determine the studied analytes.
According to the research question, it is necessary to compare a group of control pregnant women (normotensive, normal glycemia) with a group of pregnant women with GDM (normotensive and high glycemia) and also a group of pregnant women with GDM and PE (hypertensive and high glycemia), if we look at the factors that affect pregnant patients, This makes us compare the results employing a two-way ANOVA, instead of a one-way ANOVA (I deduce from the post hoc that it is a one-way ANOVA since the authors omitted the statistical test used, only mentioning the software and the post hoc used. Dear authors, please state which statistical test was used and the basis for choosing statistical methodologies. This change may
change the sense of the differences found, undoubtedly changing the meaning of discussing the results. This single observation alone leads to the manuscript being approved with major revisions.
Results section. The narration and description of the results in the text are good; it is possible to recognize the main differences and the meaning of these in the context of answering the research question and the objectives of the work set out in the manuscript. However, the figures and tables, particularly their figure and table captions, do not provide sufficient information
for the reader to understand what they show. For example,
Table 1.
*There is a letter b in the title where variables appear (BMI %). What does it mean?
*They use more and less , but in some variables shown, they appear in parentheses; what does this parenthesis mean?
*There are lines where there is only one figure (line OGTTb, which is a continuation of the text of the previous line); what does it mean?
*There are variables where the result is a threshold, less or greater than. What is the result of this determination?
Table 2.
The p value of AOPP is 0.0 ?
Figures 1, 2, 4, and 5.
The graphs do not have the significance symbols mentioned in the figure caption, and the statistical test used is not mentioned. The discussion section. It is well done, with data and references that are useful to contextualize the main findings found in this
manuscript. However, as mentioned above, the new statistical analysis could change the structure of the discussion and should be modified. In lines 198 to 200; the authors mention decreased catalase activity, but in Table 1, it appears to have increased.
There are many prolixity errors on the part of the authors of this manuscript, but it is a great subject, and the experiments performed are adequate. That is why I accept this manuscript with significant observations, and I urge the authors to incorporate my suggestions to achieve excellent quality in the final manuscript.
Author Response
Dear Reviewer,
Thank you very much for helping us to improve our manuscript, all corrections is highlighted in green.
The manuscript you submitted, "Oxidative stress and antioxidant status in pregnant women with gestational diabetes mellitus and late-onset complication of preeclampsia” attempts to evaluate oxidative and inflammatory processes in patients with gestational diabetes alone or with PE. It is an interesting line of work in an area of significant clinical impact. Unfortunately, this manuscript lacks an adequate analysis of the results obtained. Moreover, presenting the figures that show the main results must be corrected.
When reading the title and the introduction, the authors explain and develop why the study and the comparisons between the conditions of the pregnant patients are necessary. The experimental determinations are adequate to answer the questions generated and thus account for the objective of the manuscript. The introduction is agile and precise in the statement of ideas, and the title gives an account of the main aim of the manuscript.
Point 1: In the methodology section, the techniques are well described in some points; in others, they are very briefly described, for example, in sections 4.4.1 and 4.4.2, which mention modifications but do not express them. Although all the methodologies have an associated reference, should the manuscript reader understand that the same equipment was used, please mention which equipment the authors used to determine the studied analytes.
Answer 1: Done
Point 2: According to the research question, it is necessary to compare a group of control pregnant women (normotensive, normal glycemia) with a group of pregnant women with GDM (normotensive and high glycemia) and also a group of pregnant women with GDM and PE (hypertensive and high glycemia), if we look at the factors that affect pregnant patients, This makes us compare the results employing a two-way ANOVA, instead of a one-way ANOVA (I deduce from the post hoc that it is a one-way ANOVA since the authors omitted the statistical test used, only mentioning the software and the post hoc used. Dear authors, please state which statistical test was used and the basis for choosing statistical methodologies. This change may change the sense of the differences found, undoubtedly changing the meaning of discussing the results. This single observation alone leads to the manuscript being approved with major revisions.
Answer 2: Thank you for the correction; while writing the manuscript, we forgot to add a one-way ANOVA test. Regarding the two-way ANOVA, our statisticians were unable to use our data for this analysis. According to them, two-way analysis of variance (ANOVA) is an extension of one-way ANOVA, which examines the effect of two different categorical independent variables on a single continuous dependent variable.
Point 3: Results section. The narration and description of the results in the text are good; it is possible to recognize the main differences and the meaning of these in the context of answering the research question and the objectives of the work set out in the manuscript. However, the figures and tables, particularly their figure and table captions, do not provide sufficient information for the reader to understand what they show. For example, Table 1.
*There is a letter b in the title where variables appear (BMI %). What does it mean?
Answer 4: it is mistake
Point 5: *They use more and less , but in some variables shown, they appear in parentheses; what does this parenthesis mean?
Answer: we corrected it.
Point 6: *There are lines where there is only one figure (line OGTTb, which is a continuation of the text of the previous line); what does it mean?
Answer: we corrected it.
Point 7: *There are variables where the result is a threshold, less or greater than. What is the result of this determination? Table 2.
The p value of AOPP is 0.0 ?
Answer 7: done
Point 8: Figures 1, 2, 4, and 5. The graphs do not have the significance symbols mentioned in the figure caption, and the statistical test used is not mentioned.
Answer 9: done
The discussion section. It is well done, with data and references that are useful to contextualize the main findings found in this
manuscript. However, as mentioned above, the new statistical analysis could change the structure of the discussion and should be modified. In lines 198 to 200; the authors mention decreased catalase activity, but in Table 1, it appears to have increased.
There are many prolixity errors on the part of the authors of this manuscript, but it is a great subject, and the experiments performed are adequate. That is why I accept this manuscript with significant observations, and I urge the authors to incorporate my suggestions to achieve excellent quality in the final manuscript.
Reviewer 2 Report
Comments and Suggestions for Authors
Dear Authors,
You present here a study which had as theme the comparison of the oxidative stress biomarkers and of the antioxidant status in pregnant women with gestational diabetes and with preeclampsia and diabetes.
The theme of oxidative stress in pregnant women is of high interest for the pharmaceutical and medical community, but the novelty and originality of the study is very low, due to the fact that is already well-known and very logic that the oxidative stress is higher in pregnant women with gestational diabetes in comparison with normal pregnancies. Preeclampsia comes with high blood pressure, also associated with distress and oxidative stress, therefore, again, very well-known and logic, that a double threat (diabetes with preeclampsia) comes with double oxidative stress. In conclusion, I do not find something original or of novelty in your study, only the reconfirmation of facts already known.
In the manuscript, it is not clear to what you are making a reference in lines 76-78. The figures need to be more organized and easier to follow. The figures require to have a name, not a description. The title of the figures comes usually after the figure itself. In line 100, you should detail what AOPP stand for. In figure 4, there is no notation for Figure A or Figure B. Plus figures 4C and 4D are smaller than 4A and 4B. the Conclusions part need to be redone.
Author Response
Dear reviewer,
Thank you very much for helping us to improve our manuscript, all corrections is highlighted in green.
Point 1: You present here a study which had as theme the comparison of the oxidative stress biomarkers and of the antioxidant status in pregnant women with gestational diabetes and with preeclampsia and diabetes. The theme of oxidative stress in pregnant women is of high interest for the pharmaceutical and medical community, but the novelty and originality of the study is very low, due to the fact that is already well-known and very logic that the oxidative stress is higher in pregnant women with gestational diabetes in comparison with normal pregnancies. Preeclampsia comes with high blood pressure, also associated with distress and oxidative stress, therefore, again, very well-known and logic, that a double threat (diabetes with preeclampsia) comes with double oxidative stress. In conclusion, I do not find something original or of novelty in your study, only the reconfirmation of facts already known.
Answer 1: The problem of OS has been developing for years, with knowledge significantly advancing in various aspects of medical science, obstetrics and gynecology being no exception. We agree with your statement that it is known that oxidative stress is higher in pregnant women with gestational diabetes and pregnant women with gestational diabetes and preeclampsia. We confirm this with our results. In addition, our study aimed not only to reconfirm the presence of oxidative stress in the presented groups of patients but also to investigate the potential relationship between markers of oxidative stress during pregnancy, the likelihood of developing gestational diabetes, and the subsequent complication – preeclampsia with a preventive purpose in risk groups of pregnant women. For this purpose, we used an assessment of free radical levels by electron paramagnetic resonance spectroscopy, which are the most modern methodology and gold standard in redox biology and biomedicine. Based on these results, we could target at-risk groups of pregnant women for assessment of the studied OS indicators for early diagnosis and, above all, prevention of such serious, pregnancy-threatening complications.
Point 2: In the manuscript, it is not clear to what you are making a reference in lines 76-78. The figures need to be more organized and easier to follow.
Answer 2: Done
Point 3: The figures require to have a name, not a description. The title of the figures comes usually after the figure itself.
Answer 3: Done
Point 4: In line 100, you should detail what AOPP stand for.
Answer 4: Done
Point 5: In figure 4, there is no notation for Figure A or Figure B. Plus figures 4C and 4D are smaller than 4A and 4B.
Answer 5: Done
Point 6: Conclusions part need to be redone.
Answer 6: Done
Round 2
Reviewer 2 Report
Comments and Suggestions for Authors
Dear Authors,
Thank you for considering my comments and for making some of the changes suggested. Still, the figures do not have a name, for ex. Figure 4: "Figure 4. Present the NO levels...". This is not a title, it is a description of the data presented in it.
As I said the first time in my review, I do not find the elements of originality, even that you say that you used some new modern techniques for investigation.
Author Response
Comments 1: Figure 4: "Figure 4. Present the NO levels..." This is not a title; it is a description of the data presented in it.
Response 1: Done
Comments 2: As I said the first time in my review, I do not find the elements of originality, even though you say that you used some new modern techniques for investigation.
Response 2: The novelty of the present study is that it is the first comprehensive profiling of oxidative/inflammatory pathways in the comorbidity of GD and PE. It identifies potential mechanistic links between oxidative stress, GD severity, and the onset of PE. It provides a basis for novel therapeutic strategies targeting oxidative stress (e.g., antioxidant therapies) in conjunction with glycemic control.